

# Human poses recognition based on Spiking Pulse Graph Neural Networks

Shenming Qu[*], He Li[*] and Zilong Pang

Software College, Henan University, Kaifeng, Henan, China
[*] These authors contributed equally to this work.

## ABSTRACT

The temporal dilated convolutional model usually requires a large amount of computing resources in the human pose recognition task, especially when the input image information is too complex, which will lead to the shortcomings of low accuracy and large model energy consumption. Addressing these issues, we designed a new human poses recognition Spiking Pulse Graph Neural Networks model. In this model, we increase the receptive field of the model by changing the expansion coefficient and activation function of the convolution in the model receptive field processing module, so as to make the extracted features more accurate. The transmission rate of feature information is controlled by the Spiking Pulse Graph Neural Networks model with the reward mechanism of human pose learning rate, which is used to improve the accuracy of the model and reduce the energy loss of the model. Compared with the temporal dilated convolutional model and the latest human pose recognition method, the accuracy of human pose recognition is improved and the energy consumption is reduced under the same test set data.

## INTRODUCTION

In the era of digital intelligence, human poses recognition technology plays a vital role in life. Whether it is the automatic diagnosis of medical images, the anomaly detection of security monitoring systems, or the VR judging in international competitions, the application of human poses recognition can be seen everywhere in our lives. However, as image resolution increases, image data becomes more and more complex. Commonly used human poses recognition methods, such as graph convolution method and temporal dilated convolution method, all have problems such as low accuracy and high energy consumption in the process of image recognition. Among them, the problem of high energy consumption in image recognition makes these methods not well applicable to small outdoor mobile detection equipment or electric vehicle automatic driving equipment. Therefore, this paper improves on the existing temporal dilated convolutional model and designs and implements a human posture recognition method that takes into account both accuracy and energy consumption.

Corresponding author
Zilong Pang, jszxpzl@henu.edu.cn

Graph convolutional network models (GCNs) provide a lightweight solution to represent the structure of the human body in human poses recognition. *Osokin (2018)* proposed a semantic GCN, which uses encoding to represent the local and global relationships between different body joints, and realizes 2D to 3D poses regression. *Cai et al. (2019)* used spatiotemporal GCNs to capture human poses outside of structural constraints. However, the method proposed by *Osokin (2018)* will generate a lot of energy consumption in the process of encoding and decoding image eigenvalue information. The method proposed by *Cai et al. (2019)* will reduce the accuracy of image recognition in the process of capturing human nodes outside the structural constraints.

Temporal dilated convolutional have been successfully used for audio generation (*van den Oord et al., 2016*), semantic segmentation (*Yu & Koltun, 2015*), and machine translation (*Kalchbrenner et al., 2016*). *Pavllo et al. (2019)* proposed a temporal dilated convolutional to identify human poses. This model is a fully convolutional architecture with residual connections that takes a series of 2D poses as inputs and and transform them through eleven layers of temporal convolution. In a convolutional model, the gradient path between the output and input has a fixed length, regardless of the length of the sequence, which mitigates the effect of the disappearance and explosion of the gradient on the neural network. However, when using the temporal dilated convolutional model to identify human posture, the eigenvalue information is transmitted in decimal form. This information transmission method consumes a lot of energy in complex images and is difficult to deploy in restricted environments. Therefore, this article is improved on the basis of temporal dilated convolutional, retaining the advantages of time dilation convolution while reducing the recognition energy consumption of the model.

In order to solve the problem of excessive energy consumption in human poses recognition, we propose a novel solution. It is to improve the dilated coefficient of the original temporal dilated convolutional, and Spiking Pulse Graph Neural convolution algorithm module with learning rate reward mechanism processing is added to the convolution layer. As shown in Fig. 1, this paper is an improvement on the time expansion convolution method. The temporal expansion convolution originally had 11 layers of convolution with an expansion coefficient of 1, but in this paper, the parameter value of the expansion coefficient is modified to 2 starting from the 7th layer of the convolutional layer. This allows the model to increase the receptive field without losing the size of the feature map, so that the image feature value extraction is more accurate. However, due to the change of the expansion coefficient, the gap between the convolution kernels increases, resulting in a decrease in accuracy when inputting continuous image information. Therefore, after the 11-layer convolutional layer, the continuous convolution with expansion coefficients of 1, 2, and 3 is designed, and the expansion coefficient of the expansion convolution arranged continuously in three layers is zigzag-like, which solves the problem of inaccurate information omission due to the large gap between the convolution kernels, and uses the exponential linear unit (ELU) as the activation function of the three-layer convolution, based on the characteristics that the ELU itself does not produce a large gradient and can obtain negative output. It is used to solve the problem of inaccurate image information recognition due to the convolution kernel gap problem in the first 11 layers. The above

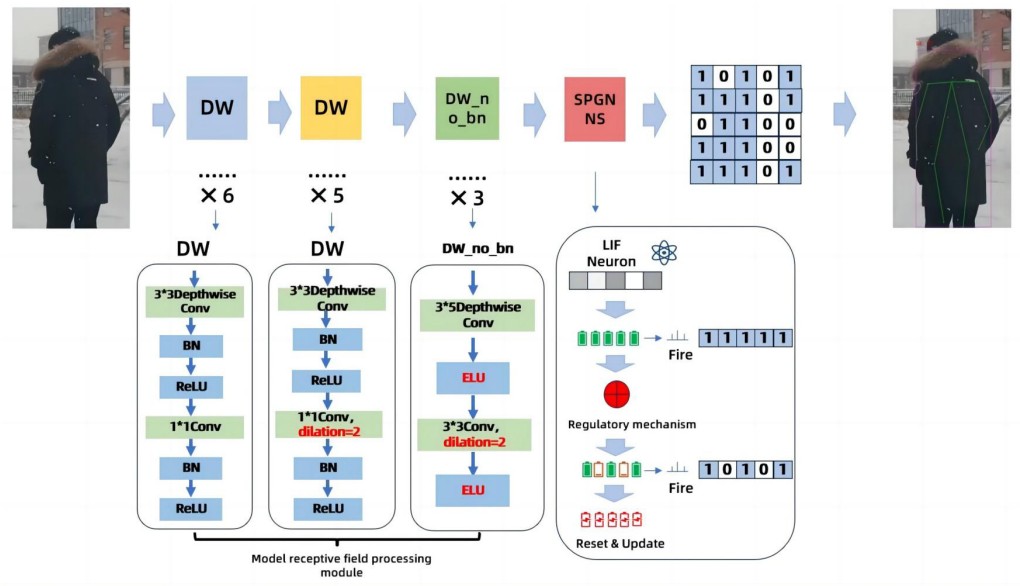

**Figure 1 Model of the Spiking Pulse Graph Neural network framework.** We set the convolution dilated parameter to 2 starting from the 7th convolution layer in order to increase the receptive field of the model. Add a module containing three layers of continuous convolution after layer 11th to better capture human poses between consecutive images. The Spiking Pulse Graph Neural convolution module with a learning rate reward mechanism was added after 14th layers to reduce the energy consumption of the model and improve the accuracy of the model.

14-layer convolution is called the model receptive field processing module. In this paper, a pulse graph convolution module with a human node recognition inhibition mechanism is added after the 14-layer convolution, in which the pulse graph convolutional layer simulates the transmission mode between neurons. And it transmits information in a binary manner. *Davies et al. (2018)*, compared with traditional deep learning decimal information transmission, this method has lower energy consumption. At the same time, in order to avoid the reduction of model recognition accuracy due to the Spiking Pulse Graph Neural Networks transmitting information too fast, we added a human node recognition suppression mechanism to the Spiking Pulse Graph Neural Networks, and the human body node recognition suppression mechanism is used to control the pulse of the Spiking Pulse Graph Neural Networks. This ensures that the method in this paper can take into account the accuracy of recognition on the basis of low energy consumption.In our experiments, we also compared the latest experimental model of *Yu et al. (2023)*, and our recognition results are better than the model of *Yu et al. (2023)* under the same data set.

## The main contributions of this paper are as follows

(1) Design of a learning rate reward mechanism, where the threshold of the membrane potential adjusts itself according to the attenuation factor with the addition of a learning rate reward mechanism, thus improving the accuracy of the model.

(2) Design the model perception module to increase the receptive of the model by changing the dilated coefficient of the convolutional layer to enhance the extraction of the image eigenvalue data.

(3) Optimization of the activation function, Exponential Linear Unit function is introduced to increase the accuracy of capturing human poses between consecutive images.

(4) A novel human poses recognition framework model, the Spiking Graph Neural Networks model (SPGNNs), is designed; this model outperforms the commonly used human gesture recognition framework models such as temporal dilated convolutional in terms of energy consumption in human poses recognition work.

## RELATED WORK

In this section, we focus on the two aspects most relevant to our work, namely temporally dilated convolutional networks and spike graph convolutional networks.

### Temporal dilated convolutional model

*Pavllo et al. (2019)* proposed a temporal dilated convolutional model to identify human poses, which solved the defect that Convolutional Neural Network (CNN) could not be refined when extracting features (*Bai, Kolter & Koltun, 2018*). Increase the receptive field by changing the dilated coefficient of the convolutional layer, ensuring that the receptive field can cover the input block information. As shown in Fig. 2A, the convolutional neural network needs to pass through two layers to obtain the historical data information of the lower layer with a 5 step size, if it is historical data information with a step size, then 50 layers of neural network are required. *Pavllo et al. (2019)* found that image information was extracted several times under this method. Therefore, a new dilated coefficient is introduced, and as long as the dilated coefficient is less than the step size, there will be no information omission (*Bai, Kolter & Koltun, 2018*). Figure 2B set the dilated coefficient to 2, so we get more information about the upper layer of the data. However, in the process of continuous image information recognition, the method proposed by *Pavllo et al. (2019)* is inaccurate due to the large gap between the convolution kernels. In this paper, the method of changing the expansion coefficient of the convolutional layer on the temporal dilated convolutional to increase the receptive field is adopted. In order to solve the problem of the accuracy of continuous image capture in *Pavllo et al. (2019)*, a module containing three consecutive convolutional layers was added. At the same time, in order to reduce the energy consumption of the model, this article adds Spiking Pulse Graph Neural Networks with a learning rate reward mechanism at the bottom of the model.

### Spiking graph neural networks

The Spiking Graph Neural Networks (*Zhu et al., 2022*; *Bacho & Chu, 2022*) is an efficient and energy-saving network. It is different from the common deep learning neural network signal transmission method. The common deep learning neural network uses continuous decimal values for signal transmission. The Spiking Graph Neural Networks is transmitted in binary mode. And it is popular because of the unique properties of

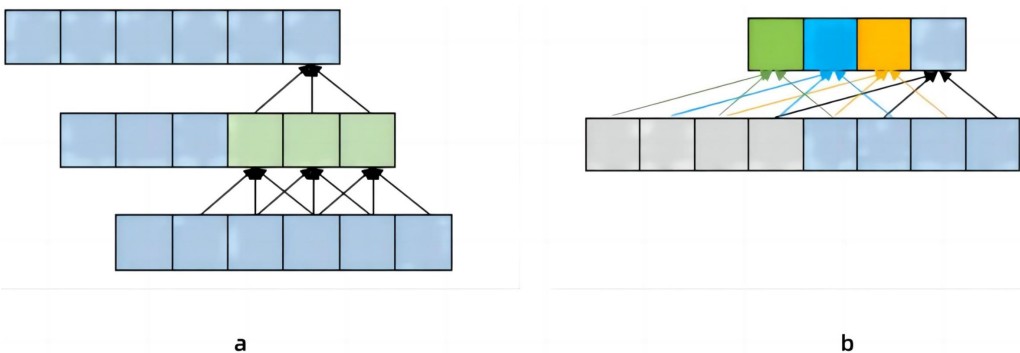

a                                                 b

**Figure 2 The contrast between (A) convolutional neural network and (B) temporal dilated convolutional information propagation.** (A) Convolutional neural network which has a convolutional kernel of 3 and a dilated factor of 1. (B) Temporal Convolutional Network, which has a convolution kernel of 3 and a dilated coefficient of 2.

low power consumption and bioreasonableness of this method (*Bu et al., 2022*). At its core, Spiking Graph Neural Networks simulate how neurons work in the human brain. The ensemble and ignition (IF) model (*Li et al., 2023*) is the mainstream computational model for simulating the way neurons work in networks. The IF model represents the membrane potential as a charge stored on a capacitor and abstracts the neuronal dynamics of a pulse-graph neural network into three temporal events: (i) Accumulation: Over time, neurons integrate currents through capacitors, resulting in charge accumulation. (ii) Launch: When the membrane potential reaches or exceeds a given threshold $V_{th}$, it emits (*i.e.,* emits a pulse). (iii) Reset: Once triggered, the membrane potential resets to a constant value of the $V_{reset} < V_{th}$ like a biological neuron (*Leukhin et al., 2020*). However, neuronal membranes are not perfect capacitors, and instead, they slowly leak current over time, pulling the membrane voltage back to its resting potential (*Lagani et al., 2023*). Therefore, the IF model needs to consider the leakage current. *Zhu et al. (2022)* added a leakage term. Neuronal activity is described as the integration of the received spike voltage and the weak dissipation to the environment, as shown in Eq. (1): where $\triangle V_m$ is the presynaptic input of the membrane voltage; $\tau_m$ is a membrane-related hyperparameter that controls the rate of decay of the membrane potential, resulting in an exponential charge and discharge of the membrane potential within the membrane. However, *Zhu et al. (2022)* only applied Spiking Graph Neural Networks to handwriting recognition. We then used Spiking Pulse Graph Neural Networks for human poses recognition for the first time, and further reduced the energy consumption of the model by learning the rate reward mechanism.

$$\tau_m \frac{d_V}{d_t} - (V - V_{reset}) + \triangle V_m. \tag{1}$$

## MovePose model

*Yu et al. (2023)* present MovePose, an optimized lightweight convolutional neural network. MovePose is designed leveraging large convolutions to expand the receptive

field, scrutinizing a broader area of feature maps, and thereby acquiring superior global features. This algorithm also uses a deconvolution network as a substitute for the usual bicubic interpolation upsampling approach, simplifying the complexity of estimation along with an increase in precision (*Yu et al., 2023*). It suitable for real-time applications such as fitness tracking, sign language interpretation, and advanced mobile human poses estimation. Therefore, in this paper, the MovePose model is selected as the comparison model of the proposed method. In the experiment, we compared the proposed Spiking Pulse Graph Neural convolution network model with the MovePose model under the Human3.6M datasets, and our recognition effect was better.

## METHODS

This section elaborates on the method we proposed. First, we introduce the receptive field processing method of the Spiking Pulse Graph Neural Networks models we proposed. Then the optimization of the activation function, the introduction of the Spiking Pulse Graph Neural Networks and the construction of the learning rate reward mechanism are introduced in turn.

### The receptive field of the model is enlarged

We segment the input video information through OpenCV to obtain the data of each frame. Then, the data of each frame is passed into the convolutional layer for feature value extraction. We improve the dilated coefficient in the temporal dilated convolutional model (*Yu, Koltun & Funkhouser, 2017*), replace the layers 7 to 11 convolutions in the model with dilated convolutions. In this paper, we first keep the expansion coefficient of the first six layers of convolution unchanged at 1, so that the filter will be applied to each pixel to improve the resolution. The expansion coefficient is then set to 2 from layer 7, which allows the convolutional layer to gradient the expansion coefficient and increase the receptive field of the convolution without losing the size of the feature map. And the gap between 2 and 1 is small, so that the method can maintain a certain level of feature resolution and capture more features in the input data. After experimental comparison, it is found that when the number of layers of dilated convolution is close to the same as the number of layers of non-dilated convolution, the recognition of human nodes in the image is relatively clear. However, this article is about capturing human nodes with a continuous set of images, which also introduces new problems. It is mainly reflected in the input of convolution, because the expansion coefficient is not 1, there is a gap between the convolution kernels, which leads to the reduction of accuracy when inputting multiple frames and continuous image information in the experiment. Therefore, in order to improve the accuracy of image information acquisition, a module containing three layers of convolution was added after the 11-layer convolution processing in the receptive field processing module. The expansion coefficients of these three layers of convolution are 1, 2, and 3 respectively, and the expansion coefficients of the three consecutive layers of expansion convolutions are arranged in a zigzag pattern. It solves the problem of inaccurate information omission due to the large gap between the convolution kernels. As shown in Fig. 3. The blue identifier is the convolution center involved in the calculation, while the color depth indicates the

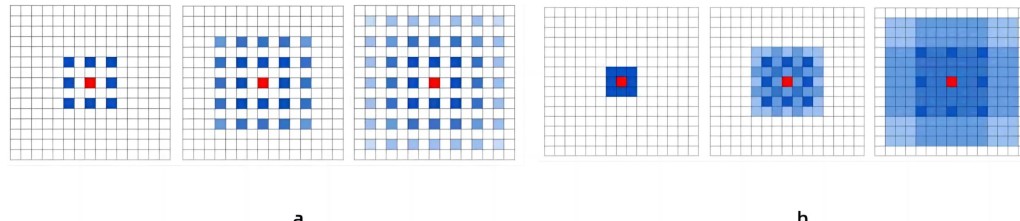

**Figure 3** Comparison of the results of the two schemes after continuous convolution of expansion convolution.

number of times the feature is extracted. It can be seen that in Fig. 3A, since the expansion coefficient of the 3rd order is the same, the computational center of the convolution will show a grid expansion outward, and some points will not become the center point of the computation. Therefore, in this paper, the expansion coefficient is set to 1, 2, 3 in the additional three-layer convolution, so that it is "jagged", then the distribution of the convolution center will become the result of Fig. 3B, and the point in the image will become the center point of the calculation, and there will be no omission.

## Activation function optimization

On the basis of the increase of the receptive field of the model, the activation function of the three-layer continuous convolution is optimized. As shown in Fig. 4, in this paper, the activation function of the first 11 layers of convolution is the Rectified Linear Unit (ReLU), and from layers 12 to 14, we replace the activation function with the ELU activation function. Because the derivative of the ELU activation function converges to zero, when the image information input is abnormal, ReLU will produce a large gradient in the backpropagation, which will lead to neuronal death and gradient disappearance. ELUs, on the other hand, do not produce large gradients and can get negative outputs, which helps the network push weights and biases in the right direction. It also prevents the emergence of dead neurons (*Wei et al., 2016*), which improves learning efficiency. At the same time, ELU has a faster recognition rate compared to ReLU.

## Spiking pulse graph neural networks

On the basis of the above, the pulse graph convolution algorithm is introduced, and the purpose is to apply the pulse convolution in binary form of data transmission to human node recognition. Because the Spiking Pulse Graph Neural Networks expression in the method of *Zhu et al. (2022)* is used for the recognition of handwritten words. In order to better apply the Spiking Pulse Graph Neural Networks to the scene of human node recognition, and ensure the availability of the Spiking Pulse Graph Neural Networks (*Fang et al., 2021*). In this paper, the differential equation in the method of *Zhu et al. (2022)* is converted into an iterative expression for human node recognition, such as Eq. (2): where $I^t$ represents the presynaptic input from the previous neuron at time step t, and $V^t$

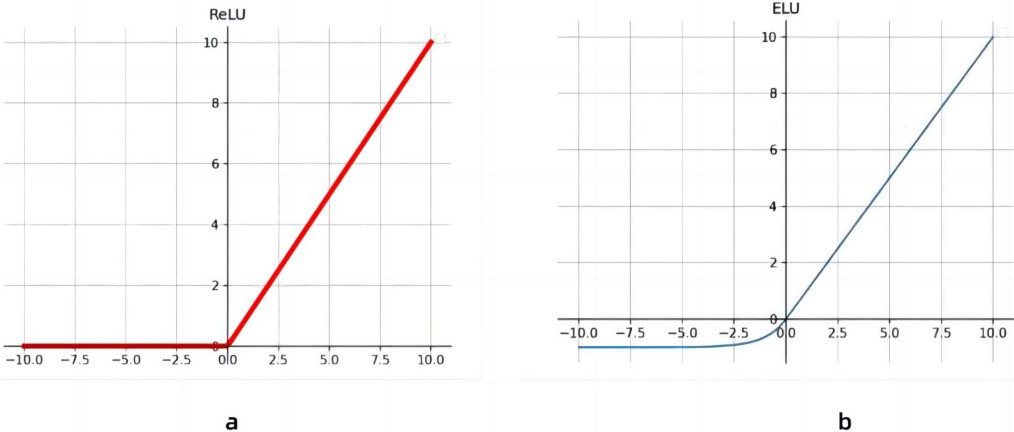

**Figure 4 Activation function linear correction function with exponential linear unit.** (A) Activation function of the Linear rectification function, (B) Exponential Linear Unit activation function.

represents the set of $N$ poses at time $t$, where $V_{reset}$ is the reset voltage.

$$V^t = V^{t-1} + \frac{1}{\tau_m}(-(V^{t-1} - V_{reset}) + I^t). \tag{2}$$

In the extended Spiking Pulse Graph Neural Networks proposed in this paper, the 18 key nodes in each frame of the image are mainly captured and the edges generated by the connection of key nodes. The 14-layer convolution in the model receptive field processing module extracts the image feature values and transmits them to the Spiking Pulse Graph Neural Networks layer. The Spiking Pulse Graph Neural Networks samples the coordinate set of key nodes, and collects the adjacent nodes of each node to aggregate pulse signals. Subsequently, the Leaky integrity-Fire (LIF) model in the Spiking Pulse Graph Neural Networks takes the aggregate signal as the input and captures the node information through an ensemble and triggering mechanism. When the key node information in the image is passed into the Spiking Pulse Graph Neural Networks,the membrane potential charge of the key node will be generated, and the charge will be stored in the capacitor. When the charge in the capacitor reaches a given threshold $V_{th}$, a pulse is made. The pulse process is transmitted by binary electrical signals, and after each pulse is triggered, the capacitance is released and the membrane potential resets to a constant $V_{reset}$. However, when too much image information is incoming, the membrane potential voltage will accumulate quickly, resulting in excessive pulse frequency and inaccurate image recognition. Therefore, this paper also adds a suppression mechanism, which inhibits the voltage accumulation rate of the Spiking Pulse Graph Neural Networks through the inhibitor $\tau_{th}$. $\tau_{th}$ can automatically change the size of the dataset after multiple rounds of training, and when the pulse frequency is too fast due to too much image data, the $\tau_{th}$ can suppress the pulse frequency by changing its own size.

**Learning rate reward mechanism**

In fact, in the Spiking Pulse Graph Neural Networks, $V^t$ is constantly accumulating and dynamically changing with the continuous transmission of data. If the voltage accumulation

rate of the membrane potential is a static value, the final result obtained in this paper is inaccurate. When the image is recognized at close range for multiple people, too much node information in the image leads to the rapid charge accumulation of the membrane potential and the number of pulses, so that many non-critical nodes appear out of thin air in the recognition results, for example, there is no leg information on the image but leg nodes are generated. Therefore, in order to solve this problem, this paper adds a suppression mechanism to the Spiking Pulse Graph Neural Networks.

First, LIF's neuronal behavior is characterized by a series of events: integration, impulse, and reset. For each frame of image, each neuron in the LIF model updates the membrane potential based on its memory state and current input, and then performs a pulse when the membrane potential reaches a threshold $V_{th}$. A pulse is triggered, the membrane potential is reset $V_{reset}$, and then the next round of the process begins again. The core of the inhibition mechanism is to suppress the rate of charge accumulation of the membrane potential by inhibiting the rate of charge accumulation through the inhibitor $\tau_{th}$ so as to suppress the frequency of pulses. As shown in Eq. (3),where $\tau_{th}$ is the inhibitor, and $\rho$ represents the vector of key nodes in the image. $Q^t$ is a pulse sequence, expressed as a vector, with a binary value (0 or 1) indicating whether the neuron is pulsed at moment $t$. $\tau_{th}$ can automatically change the size of the dataset after multiple rounds of training, and when the pulse frequency is too fast due to too much image data, the $\tau_{th}$ can suppress the pulse frequency by changing its own size. $\tau_{th}V_{th}^{(t-1)}$ represents the membrane potential voltage at time $t-1$ under the action of the inhibitor, $\rho Q^t$ represents the voltage to be accumulated at time t, and $V_{th}^t$ represents the final membrane potential voltage at time $t$.

$$V_{th}^t = \tau_{th}V_{th}^{t-1} + \rho Q^t. \tag{3}$$

As shown in Fig. 5, when the image data is too complex, the charge of the neuronal membrane potential accumulates quickly, and the number of pulses is too large, as shown in Fig. 5A, the data results obtained are very different from the input results. When the inhibition mechanism is added, the accumulation rate of the membrane potential adjusts itself according to the inhibitory factor $\tau_{th}$ which inhibits the rate at which the membrane potential stores charge and thus reduces the number of pulses. As shown in Fig. 5B, the data results are more accurate with the addition of the inhibition mechanism. Experimental verification shows that the inhibition mechanism improves the accuracy of the model.

# EXPERIMENTS AND DATASETS

## Experimental parameter configuration and Datasets

The experimental evaluation in this article was conducted on an NVIDIA GeForce RTX 3060 Laptop GPU, with a specific experimental configuration as Table 1. We used the public dataset Human3.6M in the experiment. Human3.6M is a classic human motion dataset widely used in computer vision and computer graphics research. This dataset provides precise annotations of multiple human poses and movements to facilitate human motion analysis, 3D poses estimation, and related fields (*Ionescu et al., 2014*). The Human3.6M dataset contains approximately 300,000 frames of high-resolution video that captures 15

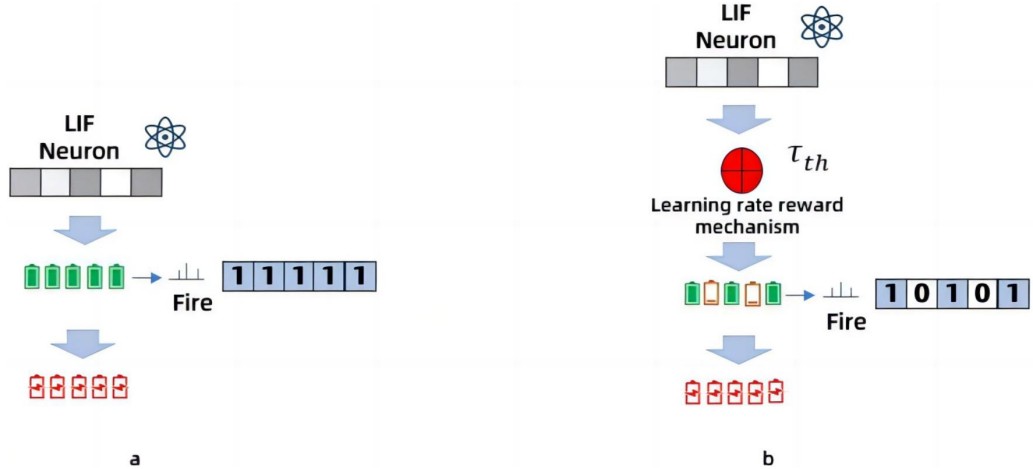

**Figure 5 Working principle of Spiking Pulse Graph Neural convolution under learning rate reward mechanism.** (A) Not processed with a learning rate reward mechanism, and the charge accumulation of neuronal membrane potential is too fast, and the data results are inaccurate. (B) Adjustment of the threshold through the attenuation factor in the learning rate reward mechanism to inhibit the rapid accumulation of charge, so as to make the data results more accurate.

**Table 1 Experimental configuration.**

| Item | CPU | Computing memory | GPU | System | CUDA | Batch size | Epoch | LearningRate |
|---|---|---|---|---|---|---|---|---|
| Content | Inteli7-11800H | 16GB | NVIDIARTX3060 | Windows11 | CUDA12.0 | 50 | 100 | 0.007 |

different actions performed by different volunteers. These videos provide rich motion information and can be used for action analysis and three-dimensional poses estimation (*Ionescu, Carreira & Sminchisescu, 2014*). The dataset also contains many different types of movements, ranging from simple standing and walking to more complex activities such as jumping, bending, pull-ups, etc. This diversity helps researchers study poses and action recognition for different types of actions. The dataset can be downloaded from the official website: https://vision.imar.ro/human3.6m/related_datasets.php.

As shown in Table 2, we divide the original dynamic data set into three major categories, each category represents a different dynamic data type. The classification of these categories is based on the content, source and characteristics of the data, and we label each data sample. The labeling process includes determining the category to which each data sample belongs and labeling each sample accordingly (*Bacho & Chu, 2023*). This step is to label the dynamic data set (*Bogo et al., 2014*). This will transform our next learning method from unsupervised learning to semi-supervised learning. Semi-supervised learning is a machine learning paradigm that deals with the situation where the training data contains both labels (known categories) and no labels (unknown categories) (*Berthelot et al., 2019*). Our purpose in utilizing this learning style is to improve model performance by combining limited label information with large amounts of unlabeled data. In the experiments in this

**Table 2  Classification table of raw data: fifteen actions of raw data into fifteen categories.**

| Category one | Category two | Category three |
| --- | --- | --- |
| baseball_pitch | bench_press | jump_rope |
| baseball_swing | clean_and_jerk | strumming_guitar |
| bowling | jumping_jacks | |
| golf_swing | pull_ups | |
| tennis_forehand | sit_ups | |
| tennis_serve | push_ups | |
| | Squats | |

paper, the performance of the proposed method is measured by accuracy (A), precision (P), recall (R), quasi-average precision (mAP), and energy consumption (E).

## Experiments
### *Modeling optimal receptive field treatment experiments*
In this paper, the expansion coefficient of the convolution from layer 7 to layer 11 is changed to increase the receptive field, so that the model can obtain more image information. In view of this method, the best effect of the model in processing images is found by changing the ratio of the number of layers of the dilated convolution to the number of layers of the non-dilated convolution. Because for the first 11 layers of convolution, the possible ratios of the expansion coefficient are 1:10, 2:9, 3:8, 4:7, 5:6, 6:5, 7:4, 8:3, 9:2, 10:1. According to the characteristics of dilated convolution, when the gap between the number of layers of dilated convolution and non-expansive convolution is too large, it cannot play an improvement role. So we rule out 1:10, 2:9, 3:8, 8:3, 9:2, 10:1. As shown in Fig. 6, experiments show that the best effect is when the ratio of the number of layers of the dilated convolution to the number of layers of the non-dilated convolution is 6:5. In other cases, the image nodes are blurred, such as the 4:7 and 7:4 cases where the key nodes of the head are not recognized. Eventually, the expansion factor was changed in the seventh layer of the model frame to ensure that the model could accept more image information.

### *Activation function optimization experiment*
In this paper, in the new three-layer convolution with expansion coefficients of 1, 2, 3, the original ReLU activation function of these three-layer convolutions is replaced with the ELU activation function to improve the accuracy of a set of continuous images capturing human body nodes. Because of the characteristics of the ELU activation function, the derivative convergence is zero, and when the image information input is abnormal, the ELU will not produce a large gradient and can get a negative output, which can help the network push the weights and biases in the right direction, and can also prevent the emergence of dead neurons, thereby improving the accuracy of recognition. As in Fig. 7, All other things being equal, we use the ReLU activation function and the ELU activation function to process the same consecutive three frames. Figure 7A shows the continuous image processed by the ReLU activation function, and Fig. 7B shows the continuous image processed by the ELU activation function. Figure 7A has a result that the human

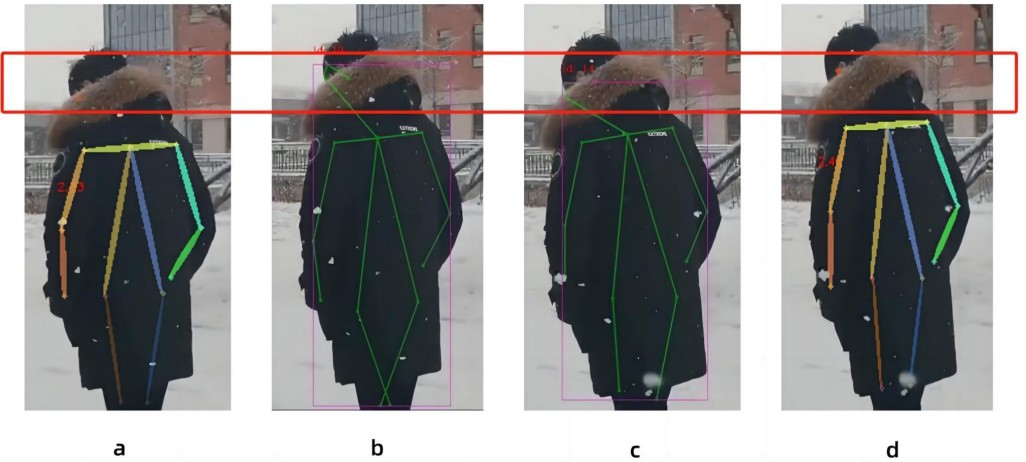

**Figure 6** Comparison of the proportion of dilated convolution layers: The four graphs are the recognition effects of different dilated convolution layers and the number of non-dilated convolution layers, (A) the effect of 4:7, (B) the effect of 5:6, (C) is the effect of 6:5, and (D) the effect of 7:4.

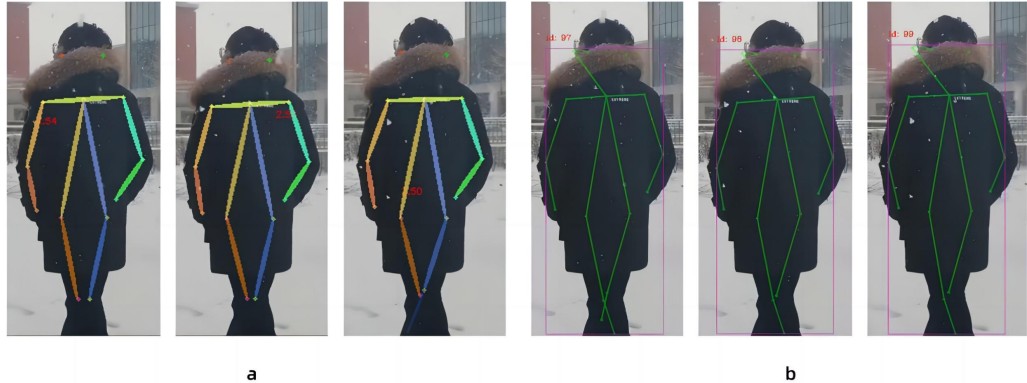

**Figure 7** Comparison between the Linear rectification function and the Exponential Linear Unit function in processing images. (A) Continuous image recognition result under the Linear rectification function, and (B) continuous image recognition result under the Exponential Linear Unit function.

body node cannot be recognized in the leg, and Fig. 7B does not have the result that the human body node cannot be recognized. According to the results, the continuous images processed under the ReLU activation function have the results that the human body nodes cannot be recognized, while the results of the continuous images processed under the ELU activation function are better than the former. Therefore, it is concluded that the human nodes between successive images are clearer when captured under ELU activation function processing.

### Experiments with learning rate incentives

In order to evaluate the performance of the proposed pulse graph neural network model with a suppression mechanism, a series of experiments are carried out. In these experiments,

**Table 3** Model comparison training results: traditional temporal dilated convolutional model approach *versus* our impulse graph neural network model trained under the same rounds respectively.

| Method | 10 | 20 | 30 | 50 | 60 | 70 |
|---|---|---|---|---|---|---|
| Temporal dilated convolution model | 0.210 | 0.221 | 0.232 | 0.287 | 0.364 | 0.291 |
| Spiking pulse graph neural networks models | 0.267 | 0.284 | 0.314 | 0.352 | 0.441 | 0.375 |

different rounds of training were carried out on the model, including 10 rounds, 20 rounds, 30 rounds, 50 rounds, 60 rounds and 70 rounds, in order to obtain the optimal inhibition mechanism $\tau_{th}$ through training. In the results, the $\tau_{th}$ for two key training rounds are obtained, which are 20 and 50 rounds, respectively. As shown in Table 3, the learning rate of the model is 0.002 in the initial stage of training. As the training progresses, the $\tau_{th}$ changes the rate of accumulation of membrane potential voltages by self-inhibiting the magnitude. When the number of training rounds exceeds 20, the learning rate accuracy of the proposed model is improved to 0.001, and the accuracy is also improved. When the number of training rounds reaches 50, the learning rate accuracy is increased to 0.0001, and the accuracy continues to improve. However, as we continued to train, we found that when the model was trained for 70 rounds, the model accuracy results decreased. As shown in Table 3, through the comparison of the final results, we found the optimal inhibitor $\tau_{60}$ within 70 rounds. However, in this paper, only less than 70 rounds of experimental training were conducted, and no more than 70 rounds of training were performed for $\tau_{th}$. Therefore, in the future, we will also conduct experiments with more than 70 rounds of training for $\tau_{th}$.

In this paper, the same video is used as the input data source, as shown in Fig. 8, the left side of the figure is the image result processed by the Spiking Pulse Graph Neural Networks without the human node recognition learning rate incentives, and the right side is the image result processed by the Spiking Pulse Graph Neural Networks with the human node recognition learning rate incentives. It can be seen that the method proposed in this paper works better at the curved overlapping parts. The Spiking Pulse Graph Neural Networks with the human node recognition learning rate incentives obtains better human node recognition results than the Spiking Pulse Graph Neural Networks without the human node recognition learning rate incentives. At the same joint site, the method in this paper can identify the nodes of the coincident site, which benefits from the influence of the inhibition mechanism $\tau_{th}$ in the method in this paper. By suppressing the rate of charge accumulation of membrane potential voltage, the number of pulses is controlled, so as to accurately identify the position of the human body node in the image, and better judge and identify the node in the image.

### Ablation experiment

In this paper, the human node recognition based on the Spiking Pulse Graph Neural Networks is based on the time expansion convolution, and the model receptive field processing module and the Spiking Pulse Graph Neural Networks with suppression mechanism are added. In the model perception processing module, two innovative designs are carried out, which are to change the expansion coefficient of the model and increase

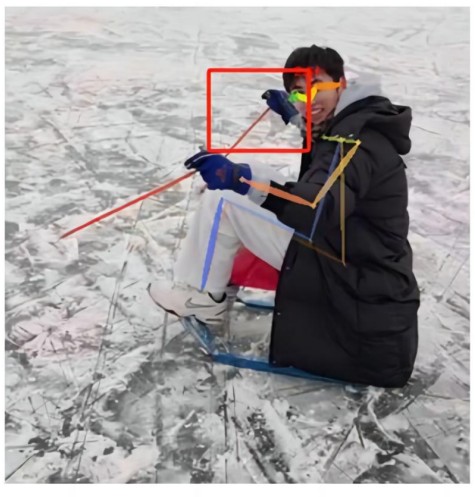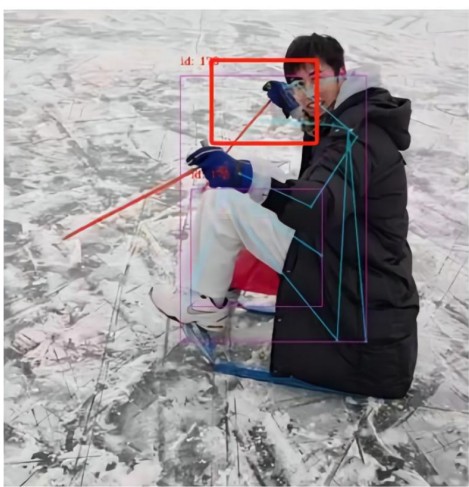

**a**                                              **b**

**Figure 8** **Comparison of experiments without the addition of inhibition mechanism and with the addition of inhibition mechanism.**

**Table 4** **Ablation experiment results.**

| Rounds | Datasets | A | P | R | mAP | E |
|---|---|---|---|---|---|---|
| Temporal dilated convolution (TDC) | Human3.6M | 0.694 | 0.694 | 0.612 | 0.424 | 984J |
| TDC+coefficient of expansion | Human3.6M | 0.701 | 0.712 | 0.756 | 0.538 | 821J |
| TDC+ receptive field processing module | Human3.6M | 0.762 | 0.762 | 0.743 | 0.566 | 812J |
| TDC+ receptive field processing module+ spiking pulse graph neural networks | Human3.6M | 0.812 | 0.821 | 0.783 | 0.652 | 432J |
| Ours | Human3.6M | 0.834 | 0.831 | 0.794 | 0.662 | 430J |

the continuous three-layer convolution with ELU activation function. Two innovative designs are also carried out in the Spiking Pulse Graph Neural Networks with inhibition mechanism, namely, the iterative Spiking Pulse Graph Neural Networks to human node recognition and the design of human node recognition inhibition mechanism.

As shown in Table 4, this article conducts ablation experiments on these four points. From the results, it can be seen that the method mentioned in this article reduces the energy consumption during the human node identification process, and is also better than time dilation convolution in terms of accuracy. Moreover, through ablation experiments, it is concluded that the several works we have done have improved the performance of the method to varying degrees.

### Model comparison experiments

In this paper, the accuracy and energy consumption of the temporal dilated convolutional model, the Spiking Graph Neural Networks, the MovePose method and we proposed in this paper are compared on the Human3.6M dataset. As shown in Fig. 9, the proposed model can identify human nodes more clearly by comparing the output results. Compared

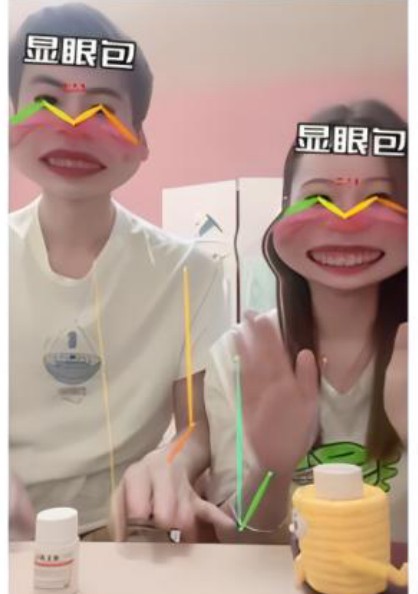
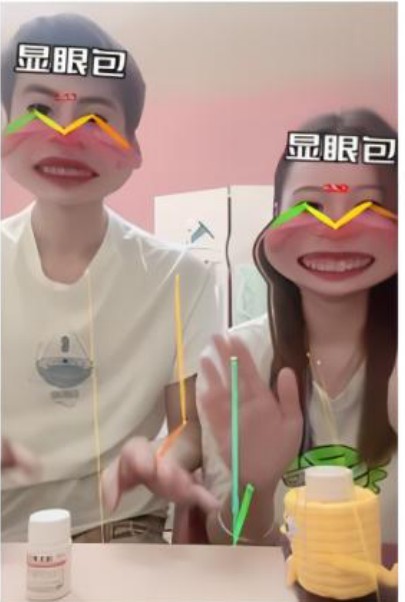

a

b

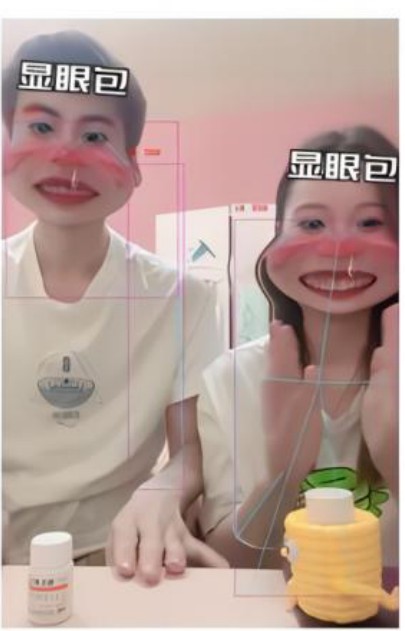
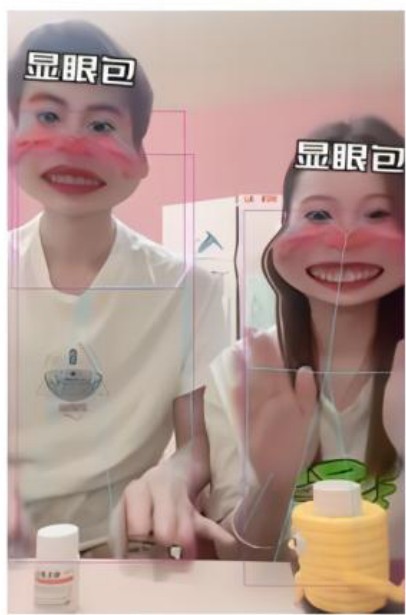

c

d

**Figure 9** **Experimental results of different methods and the methods presented here.** (A) Recognition result of the temporal dilated convolutional model, (B) Spiking Graph Neural Networks, (C) recognition result of the MovePose method, and (D) recognition result of the proposed method.

**Table 5  Ablation experiment results.**

| Rounds | Datasets | A | P | R | mAP | E |
|---|---|---|---|---|---|---|
| Temporal dilated convolutional model | Human3.6M | 0.694 | 0.694 | 0.612 | 0.424 | 984J |
| Spiking graph neural networks | Human3.6M | 0.812 | 0.821 | 0.783 | 0.652 | 432J |
| MovePose | Human3.6M | 0.822 | 0.831 | 0.792 | 0.651 | 749J |
| Ours | Human3.6M | 0.834 | 0.831 | 0.794 | 0.662 | 430J |

with the results of the convolutional neural network method of the Fig. 9B Spiking Graph Neural Networks and we proposed in this paper, the node recognition of Fig. 9D is clearer and more accurate because of the inhibition mechanism of human node recognition. At the same time, the accuracy of the proposed method is higher than that of the MovePose method compared with the result Fig. 9C of the MovePose method.

As shown in Table 5, on the Human3.6M dataset, we proposed in this paper shows low energy consumption and high accuracy in the training and inference process. This also confirms that Spiking Graph Neural Networks can reduce energy consumption while ensuring accuracy in specific target recognition applications. It provides a new solution for future target-specific identification applications.

## CONCLUSION

In this paper, a human body node recognition method based on Spiking Pulse Graph Neural Networks is proposed, and a model receptive field processing module is designed and added to the temporal dilated convolutional model, and a Spiking Pulse Graph Neural Networks algorithm module with human body node recognition reward mechanism is added to the convolutional layer. The incoming data is transmitted in binary form, which reduces the energy consumption of the model to process human node recognition. The frequency of pulses is controlled by controlling the accumulation rate of membrane potential voltage through the reward mechanism, so as to improve the accuracy of the model. In this chapter, the training and testing of the Human3.6M dataset proves that the Spiking Pulse Graph Neural Networks also has a certain effect on human node recognition in target recognition. This human body node recognition method of Spiking Pulse Graph Neural Networks can significantly reduce energy consumption while maintaining good human body node recognition accuracy. Compared with the comparison method, the energy consumption is reduced by 42.6%, and the accuracy is improved by 1.2%. It is verified that theSpiking Pulse Graph Neural Networks can reduce the energy consumption while ensuring the accuracy of the specific target recognition scene in target recognition. A new scheme for human node recognition in specific target recognition is proposed.

This result has important implications for deploying deep learning models in resource-constrained environments. The design of our proposed Spiking Pulse Graph Neural Networks allows it to utilize computational resources more efficiently and reduce energy consumption, thus making it ideal for processing dynamic and image data. It also verifies that the introduction of our proposed learning rate reward mechanism can improve the efficiency and performance of the model while reducing the energy consumption. However,

the overfitting problem for models larger than 100 rounds of training still requires more in-depth research and solutions. In the follow-up work, we will further optimize the model and explore more training strategies to improve the effectiveness of the semi-supervised learning task. In the future, we will also continue to apply the Spiking Pulse Graph Neural Networks to other recognition application scenarios.

## ACKNOWLEDGEMENTS

The authors would like to thank all the anonymous reviewers for their helpful comments and suggestions to improve the manuscript.

### Funding

This work was supported by the Heinan Science and Technology Development Plan Project (242102210164) and the National Natural Science Foundation of China (12201185). There was no additional external funding received for this study. participated in the study design, data collection and analysis, decision to publish, and preparation of manuscript.

### Grant Disclosures

The following grant information was disclosed by the authors:
Henan Science and Technology Development Plan Project: 242102210064.
National Natural Science Foundation of China: 12201185.

### Competing Interests

The authors declare there are no competing interests.

### Author Contributions

- Shenming Qu conceived and designed the experiments, performed the experiments, analyzed the data, performed the computation work, prepared figures and/or tables, authored or reviewed drafts of the article, and approved the final draft.
- He Li conceived and designed the experiments, performed the experiments, analyzed the data, performed the computation work, prepared figures and/or tables, authored or reviewed drafts of the article, and approved the final draft.
- Zilong Pang performed the experiments, prepared figures and/or tables, authored or reviewed drafts of the article, and approved the final draft.

### Data Availability

   The training code and experimental data are available in the Supplementary File. The full dataset is available at http://vision.imar.ro/human3.6m/related_datasets.php.

### Supplemental Information

Supplemental information for this article can be found online at http://dx.doi.org/10.7717/peerj-cs.2304#supplemental-information.

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
