# Peer review of "Human poses recognition based on Spiking Pulse Graph Neural Networks"

_PeerJ Computer Science, doi:10.7717/peerj-cs.2304_

## Round 0.1 · original submission · Major Revisions

Based on the reviewers comments, the manuscript must be revised.

**Language Note:** The review process has identified that the English language must be improved. PeerJ can provide language editing services - please contact us at [email protected] for pricing (be sure to provide your manuscript number and title). Alternatively, you should make your own arrangements to improve the language quality and provide details in your response letter. – PeerJ Staff

Reviewer 1 ·

Basic reporting

The manuscript entitled “Human poses recognition based on Spiking Pulse Graph Neural Networks” has been investigated in detail. The paper proposes a new approach for human pose recognition in resource-constrained environments by introducing a Spiking Pulse Graph Neural Network model. The model improves upon the temporal dilated convolutional approach by enhancing the dilated coefficient and integrating a learning rate reward mechanism. Experimental results suggest better performance in terms of energy consumption and accuracy compared to existing methods. However, the paper lacks clarity in explaining the proposed modifications and fails to provide sufficient experimental details and analysis to support its claims. There are some points that need further clarification and improvement:
1) The introduction lacks clarity in explaining the problem statement and the significance of the proposed model.
2) The paper jumps into technical details without adequately setting up the context or explaining the motivation behind the proposed approach.
3) The structure of the paper needs improvement to ensure a logical flow of ideas from introduction to conclusion.
4) The paper mentions improvements in the dilated coefficient and the addition of a Spiking Pulse Graph Neural Network without providing sufficient explanation or justification for these modifications.
5) There is a lack of clarity in how the learning rate reward mechanism works and its specific impact on the model's performance.

Experimental design

6) The paper lacks detailed explanations of the experimental setup, dataset used, and evaluation metrics employed.
7) While the paper claims superiority over the temporal dilated convolutional model in terms of energy consumption and accuracy, the experimental results are not adequately presented or analyzed.
8) The absence of comparative analysis with existing methods and benchmarks limits the credibility of the claimed improvements.

Validity of the findings

9) “Discussion” section should be added in a more highlighting, argumentative way. The author should analysis the reason why the tested results is achieved.
10) The authors should clearly emphasize the contribution of the study. Please note that the up-to-date of references will contribute to the up-to-date of your manuscript. The study named- “Overcoming nonlinear dynamics in diabetic retinopathy classification: a robust AI-based model with chaotic swarm intelligence optimization and recurrent long short-term memory”- can be used to explain the methodology in the study or to indicate the contribution in the “Introduction” section.
11) The language used is often unclear, making it difficult to understand the proposed model and its contributions.
12) The paper contains grammatical errors and lacks consistency in terminology and notation usage.
13) It will be helpful to the readers if some discussions about insight of the main results are added as Remarks.
This study may be proposed for publication if it is addressed in the specified problems.

Additional comments

As above

Cite this review as

Reviewer 2 ·

Basic reporting

Well written paper. Easy to follow and with good background material.

Experimental design

Authors focus the images on a baseball scene. I would like to see on more challenging scenarios, such as in a gym with different perspectives. Current approaches typically fail on side view of the subjects.

Validity of the findings

Authors provide source code and link to dataset. However, in the paper the address is not visible, authors must consider that the paper is static and cannot be "clicked".

Cite this review as

Reviewer 3 ·

Basic reporting

In this paper, the authors propose a new Spiking Pulse Graph Neural Network framework based on a learning rate reward mechanism. The work is interesting as it aims to demonstrate that the proposed framework is more efficient in terms of energy consumption than previous work, namely in image analysis for recognizing people's poses.
There are, however, situations that need to be clarified, namely:
1. the images are of low quality, with the details that would be relevant being almost imperceptible;
2. There are also some writing errors, line 42 "" and and", define the acronym LIF in the first occurrence (line 179), figure 4 - a. and b. -> (a) and (b), line 312 "human36m" -> "Human3.6M".

Experimental design

The algorithm that was used is mentioned in the "Spiking Pulse Graph Neural Networks" section. It is important to include the algorithm and how it differs from algorithms taken as a priori reference.
3. Table 2 appears to be wrong, or the text in the previous paragraphs is not coherent with the data in the table. At this point it is important to explain, if possible, what is behind the decrease in accuracy when the number of rounds increases to 70.

Validity of the findings

The conclusions are poor, without explaining the values or mentioning the improvements in terms of quantification (percentages) of the improvement in terms of energy consumption efficiency that the proposal presented presents in relation to previous works, included in the Related Work section . It is suggested that a comparative table be included.

Additional comments

No comments.

Cite this review as

---

## Round 0.2 · accepted · Accept

Dear authors,

The comments are well addressed and the manuscript can be accepted as it is.

Reviewer 1 ·

Basic reporting

My comments have been addressed. It is acceptable in the present form.

Experimental design

My comments have been addressed. It is acceptable in the present form.

Validity of the findings

My comments have been addressed. It is acceptable in the present form.

Cite this review as